# Hydrophilic Glycoproteins of an Edible Green Alga *Capsosiphon fulvescens* Prevent Aging-Induced Spatial Memory Impairment by Suppressing GSK-3β-Mediated ER Stress in Dorsal Hippocampus

**DOI:** 10.3390/md17030168

**Published:** 2019-03-15

**Authors:** Jeong Hwan Oh, Taek-Jeong Nam

**Affiliations:** Institute of Fisheries Sciences, Pukyong National University, Busan 46041, Korea; ojhwan55@pknu.ac.kr

**Keywords:** aging, *Capsosiphon fulvescens*, cognitive dysfunction, endoplasmic reticulum stress, glycoproteins

## Abstract

Endoplasmic reticulum (ER) stress is involved in various neurodegenerative disorders. We previously found that *Capsosiphon fulvescens* (*C. fulvescens*) crude proteins enhance spatial memory by increasing the expression of brain-derived neurotrophic factor (BDNF) in rat dorsal hippocampus. The present study investigated whether the chronic oral administration of hydrophilic *C. fulvescens* glycoproteins (Cf-hGP) reduces aging-induced cognitive dysfunction by regulating ER stress in the dorsal hippocampus. The oral administration of Cf-hGP (15 mg/kg/day) for four weeks attenuated the aging-induced increase in ER stress response protein glucose-regulated protein 78 (GRP78) in the synaptosome of the dorsal hippocampus; this was attenuated by the function-blocking anti-BDNF antibody (1 μg/μL) and a matrix metallopeptidase 9 inhibitor 1 (5 μM). Aging-induced GRP78 expression was associated with glycogen synthase kinase-3 beta (GSK-3β) (Tyr216)-mediated c-Jun N-terminal kinase phosphorylation, which was downregulated upon Cf-hGP administration. The Cf-hGP-induced increase in GSK-3β (Ser9) phosphorylation was downregulated by inhibiting tyrosine receptor kinase B and extracellular signal-regulated kinase (ERK)1/2 with cyclotraxin-B (200 nM) and SL327 (10 μM), respectively. Cf-hGP administration or the inhibition of ER stress with salubrinal (1 mg/kg, i.p.) significantly decreased aging-induced spatial memory impairment. These findings suggest that the activation of the synaptosomal BDNF-ERK1/2 signaling in the dorsal hippocampus by Cf-hGP attenuates age-dependent ER stress-induced cognitive dysfunction.

## 1. Introduction

Aging is a natural process associated with functional and cognitive decline that is closely associated with neurobiological changes in the hippocampus. Indeed, the number of newborn neurons in the subgranular zone of the hippocampus declines [1,2,3]. During aging, reduction in volume and vessel degeneration in the hippocampus may also contribute to reduced neurogenesis, resulting in cognitive dysfunction [4,5]. These age-associated changes are closely linked to the proteostasis network.

The endoplasmic reticulum (ER) is a key component of the proteostasis network, and an important organelle that plays critical roles in the correct synthesis, folding, and modification of proteins, as well as in intracellular calcium homeostasis [6,7]. Aging-induced protein damage and alterations in the redox status can decrease protein folding capacity, which causes the accumulation of misfolded proteins in the ER lumen and the activation of several signaling pathways, which is known as the ER stress response [6,8,9,10]. Glucose-regulated protein 78 (GRP78), a member of the heat shock protein family localized in the ER lumen, is a marker of ER stress [11]. During ER stress, GRP78 binds to unfolded proteins and activates a multi-chaperon complex, resulting in increased ER protein-folding capacity [12]. However, severe and long-lasting ER stress, including aging-induced ER stress, causes the accumulation of unfolded or misfolded proteins and physiological dysfunction.

Brain-derived neurotrophic factor (BDNF) is a member of the neurotrophin family and plays a critical role in hippocampal-dependent learning and memory by enhancing neuronal plasticity and long-term potentiation [13,14,15,16]. Specifically, the regulation of BDNF in the hippocampus is essential for cognitive function, which involves calcium-dependent signaling mediated by protein kinases [17,18]. Recent evidence has indicated that BDNF deficiency reduces neuroplasticity and leads to spatial memory impairment in aging rat hippocampi [19,20], and impaired regulation of BDNF has been implicated in aging-associated psychiatric disorders [21,22,23]. Several neurotrophic factors, such as fibroblast growth factor-2, BDNF, vascular endothelial growth factor, and nerve growth factor, may also contribute to age-dependent deficits in hippocampal neurogenesis [19,24,25,26]. Although the decline of hippocampal neurogenesis with age cannot be explained by only one factor, these data suggest that aging-induced memory decline could be regulated by the activation of BDNF-mediated signaling in the hippocampus.

Filamentous green alga, *Capsosiphon fulvescens* (*C. fulvescens*), is widely distributed and consumed as a nutrient source due to its beneficial health effects [27,28]. Extracts from *C. fulvescens* contain various antioxidants, including polysaccharides that induce the proliferation of rat small intestinal epithelial cells, apoptosis in gastric cancer cells, and a decrease in rat cholesterol levels [29,30,31]. In addition, bioactive proteins from *C. fulvescens* decrease hydrogen peroxide-induced oxidative stress in liver HepG2 cells and inhibit the proliferation of human gastric cancer cells [32,33]. A recent study also showed that hydrophilic crude protein from *C. fulvescens* increases mature BDNF expression in the rat dorsal hippocampus and enhances spatial memory in young rats [34]. However, the effects of *C. fulvescens* glycoproteins on aging-induced cognitive dysfunction remain unclear, because the studies focusing on their beneficial effects have been mainly performed using in vitro techniques without exploring the aging brain in vivo.

In this study, we investigated whether hydrophilic glycoproteins from *C. fulvescens* (Cf-hGP) prevent the impairment in spatial memory induced by aging via regulating ER stress. First, we determined whether oral administration of Cf-hGP downregulates ER stress, which is involved in the upregulation of BDNF and matrix metallopeptidase 9 (MMP9) expression in the rat dorsal hippocampus. Next, we determined whether the Cf-hGP-induced decrease in aging-induced ER stress is associated with BDNF–tyrosine receptor kinase B (TrkB)–extracellular signal-regulated kinase (ERK)1/2 signaling. Finally, using an in vivo behavioral study, we determined whether Cf-hGP attenuates the memory dysfunction associated with aging-induced ER stress in the dorsal hippocampus.

## 2. Results

### 2.1. The Aging-Induced Increase in GRP78 Expression in the Dorsal Hippocampus Was Downregulated by Chronic Cf-hGP Administration

To investigate the neuroprotective effects of Cf-hGP against ER stress-mediated cognitive dysfunction with age, we measured the effect of chronic oral administration of Cf-hGP on ER stress. As shown by western blot in Figure 1A, GRP78 expression was significantly increased in old rats (12 months old) compared with young rats (10 weeks old) and was downregulated by chronic Cf-hGP (15 mg/kg/day) administration for four weeks. Immunofluorescence levels indicating GRP78 expression were also increased in aged dorsal hippocampi, which was attenuated by the Cf-hGP oral administration (Figure 1B).

### 2.2. Mature BDNF and MMP9 Levels Were Upregulated whereas the Aging-Induced Increase in GRP78 Expression Was Downregulated by Chronic Cf-hGP Administration

Next, we determined whether BDNF-mediated signaling is associated with the regulation of aging-induced ER stress by Cf-hGP. As shown Figure 2A,B, mature BDNF and MMP9 expressions in the dorsal hippocampi were significantly decreased with age, which was prevented and restored by Cf-hGP administration for four weeks. To further investigate whether the upregulation of mature BDNF and MMP9 expression by Cf-hGP administration attenuates aging-induced ER stress in dorsal hippocampi, anti-BNDF antibody (1 μg/μL) or MMP9 inhibitor 1 (5 μM) was infused near the dentate gyrus of the dorsal hippocampus in freely moving rats 30 min prior to the last oral administration of Cf-hGP; rats were decapitated 150 min after the last administration. The inhibition of BDNF or MMP9 signaling downregulated the Cf-hGP-induced decrease in GRP78 expression in the dorsal hippocampus (Figure 2C).

### 2.3. The Aging-Induced Decrease in GSK-3β (Ser9) Phosphorylation and the Increase in GSK-3β (Tyr216) and JNK Phosphorylation Were Downregulated by Chronic Cf-hGP Administration

Since the upregulation of mature BDNF and MMP9 expression by Cf-hGP was associated with a decrease in aging-induced ER stress, we further determined whether glycogen synthase kinase-3 beta (GSK-3β) and c-Jun N-terminal kinase (JNK), which are closely associated with ER stress, are regulated by chronic Cf-hGP administration. As shown in Figure 3, the phosphorylation of GSK-3β (Tyr216; active form) and JNK in the dorsal hippocampus was significantly increased in old rats (12 months old) compared with the young rats (10 weeks old), which was decreased by Cf-hGP administration. In particular, the aging-induced decrease in the phosphorylation of GSK-3β (Ser9, inactive form) was attenuated by the Cf-hGP administration.

### 2.4. The Increase in GRP78 Expression Caused by Aging-Induced GSK-3β/JNK Signaling Was Attenuated by Chronic Cf-hGP Administration

To determine whether GSK-3β and JNK signaling regulate the aging-induced increase in GRP78 expression in the dorsal hippocampus, we further investigated: (1) the interaction between GSK-3β and JNK signaling, and (2) the relationship between GSK-3β-mediated JNK signaling and GRP78 expression. As shown in Figure 4A,B, GSK-3β inhibition with CHIR (3 μM) significantly downregulated the aging-induced increase in JNK phosphorylation, but the phosphorylation of GSK-3β (Tyr216) was unchanged by the inhibition of JNK with SP600125 (10 µM). In addition, the inhibition of GSK-3β and JNK with CHIR (3 μM) or SP600125 (10 µM), respectively, significantly decreased the aging-induced increase in GRP78 expression (Figure 4C).

### 2.5. The Increase in GSK-3β (Ser9) Phosphorylation by Cf-hGP Was Downregulated by Inhibition of the TrkB and ERK1/2 Signaling

As GSK-3β/JNK signaling contributed to the aging-induced increase in GRP78 expression in the dorsal hippocampus, we further investigated whether the alteration of GSK-3β phosphorylation by Cf-hGP administration is associated with BDNF-dependent TrkB-ERK1/2. As shown in Figure 5, the phosphorylation of GSK-3β (Ser9) was significantly decreased with age, which was prevented with Cf-hGP administration. Moreover, the Cf-hGP-induced increase in GSK-3β (Ser9) phosphorylation was downregulated by inhibiting the TrkB and ERK1/2 with cyclotraxin-B (200 nM) and SL327 (10 μM), respectively.

### 2.6. The Aging-Induced Spatial Memory Impairment Was Downregulated by Chronic Cf-hGP Administration

Finally, we determined whether the aging-induced cognitive dysfunction is regulated by chronic Cf-hGP administration. Following the oral administration of Cf-hGP (15 mg/kg) once a day for four weeks, spatial memory was measured using the Morris water maze. In addition, the relevance between ER stress and the aging-induced decrease in spatial memory was determined by inhibiting ER stress in the dorsal hippocampus with salubrinal (1 mg/kg, i.p.) 60 min prior to acquisition training (Figure 6A). After four weeks, acquisition training was performed. On day 3 of acquisition training, the latency to reach the platform decreased up to ˂30 s (Figure 6B). Thus, seven days after withdrawal, reference memory was assessed again. The latency to reach the platform was significantly decreased by Cf-hGP administration or the inhibition of ER stress with salubrinal (1 mg/kg, i.p.) (Figure 6C). The aging-induced decrease in the frequency of crossing the platform was also attenuated by Cf-hGP administration or the inhibition of ER stress (Figure 6D).

## 3. Discussion

Aging causes cognitive decline, such as in spatial memory, as well as physiological dysfunctions. This study was designed to investigate whether Cf-hGP administration attenuates aging-induced cognitive dysfunction and its underlying mechanism of action. The chronic oral administration of Cf-hGP for four weeks significantly decreased aging-induced ER stress caused by GSK-3β (Tyr216)-mediated JNK signaling through the activation of synaptosomal MMP9–BDNF–ERK1/2 signal cascade in the dorsal hippocampus. In addition, an ER stress-induced impairment of spatial memory with age was significantly downregulated by the inhibition of ER stress or chronic Cf-hGP administration. These finding suggest that aging-induced cognitive impairment is associated with ER stress caused by GSK-3β (Tyr216)-mediated JNK signaling, which could be restored by the Cf-hGP-induced activation of synaptic BDNF-ERK1/2 signaling.

ER stress with aging induces the dysregulation of proteostasis, which can lead to the modulation of synaptic activity and cognitive impairment [7,35]. Thus, dysregulation of the ER stress response could be one of the major risk factors for neurodegenerative diseases. In this study, the ER stress marker, GRP78, was significantly overexpressed in aging rat dorsal hippocampi compared to young rats; this effect was decreased by chronic Cf-hGP administration (Figure 1A). The immunofluorescence levels of GRP78 in dorsal hippocampi were also significantly downregulated by Cf-hGP administration (Figure 1B). The dorsal hippocampus is closely associated with cognitive function, such as spatial memory [36], and in particular, the regulation of protein synthesis in the hippocampus is essential for long-term memory formation and maintenance [37]. These data indicate that the cognitive impairment that occurs with aging could result from increased ER stress in the dorsal hippocampus, because ER stress causes the dysregulation of protein synthesis. This suggests that Cf-hGP administration could downregulate aging-induced cognitive dysfunction by regulating the ER stress response in the dorsal hippocampus.

As shown Figure 2A,B, chronic Cf-hGP administration significantly downregulated the aging-induced decrease in mature BDNF and MMP9 expression in the dorsal hippocampus. BNDF is essential for synaptic plasticity, which is associated with the persistence of long-term memory by enhancing neuronal protein synthesis and the regulation of ER stress in the hippocampus [34,37,38]. In addition, MMP9 plays an important role in controlling synaptic plasticity by the extracellular cleavage of released proteins, and converts pro-BDNF to mature BDNF, which is essential for hippocampal long-term potentiation and memory [39,40]. Since extracellular mature BDNF could be regulated by MMP9, these results suggest that MMP9-mediated BDNF signaling plays a critical role in the Cf-hGP-induced decrease in ER stress-mediated cognitive dysfunction that occurs with age. In this study, the Cf-hGP-induced decrease in GRP78 expression was abolished with an anti-BDNF antibody (1 μg/μL) or through the inhibition of MMP9 activity with CHIR (10 μM) (Figure 2C). These data suggest that aging-induced ER stress is regulated through the activation of MMP9-mediated BDNF signaling by the administration of Cf-hGP.

Next, we investigated whether the aging-induced increase in ER stress in the dorsal hippocampus is associated with GSK-3β and JNK phosphorylation. GSK-3β is the most commonly expressed isoform among GSK kinases in the brain, and plays essential roles in neuronal polarity and disorders [41,42]. Furthermore, its activity is controlled by numerous signaling pathways, including tyrosine and serine/threonine phosphorylation [43,44], the phosphorylation of tyrosine 216 residue (Tyr216) leads to GSK-3β activation, whereas the phosphorylation of serine 9 residue (Ser9) reduces its activity. In addition, the pharmacological inhibition of JNK signaling attenuates neuronal excitotoxicity and neurodegenerative diseases [45,46,47]. As shown in Figure 3, the phosphorylation of GSK-3β (Tyr216) and JNK was significantly increased, and GSK-3β (Ser9) phosphorylation was significantly decreased with age; both of these effects were attenuated by Cf-hGP administration. Moreover, the phosphorylation of JNK was significantly downregulated by the inhibition of GSK-3β, but JNK did not affect the phosphorylation of GSK-3β (Tyr216) (Figure 4A,B). The inhibition of GSK-3β and JNK signaling significantly decreased the aging-induced increase in GRP78 expression in the dorsal hippocampus (Figure 4C). These data indicate that the aging-induced increase in GRP78 expression in the dorsal hippocampus is regulated by GSK-3β-mediated JNK signaling, and that Cf-hGP treatment attenuates this response through the GSK-3β-mediated JNK signaling pathway.

Since the oral administration of Cf-hGP significantly downregulated the aging-induced decrease in BDNF expression and GSK-3β (Ser9) phosphorylation, we further investigated whether the alteration of GSK-3β phosphorylation (Ser9) by Cf-hGP results from the activation of BDNF/TrkB-mediated ERK1/2 signaling. BDNF plays a critical role in synaptic plasticity, which is associated with memory processing and the persistence of long-term memory [38,48]. BDNF/TrkB signaling is essential for long-term potentiation and memory formation in the hippocampus and plays a direct role in synapse number and spine density in pyramidal neurons within the hippocampus, which is mediated by ERK1/2 activation [49,50]. In addition, GSK-3β phosphorylation (Ser9) could be regulated by ERK1/2 [51]. As shown in Figure 5, the Cf-hGP-induced increase in GSK-3β phosphorylation (Ser9) was significantly downregulated by the inhibition of the TrkB and ERK1/2 with cyclotraxin-B (200 nM) and SL327 (10 μM), respectively. These data demonstrate that the increase in GRP78 expression that occurs with age is downregulated by the chronic Cf-hGP-induced activation of synaptic BDNF-ERK1/2 signaling, which suggests that age-associated/ER stress-induced cognitive dysfunction is regulated by Cf-hGP treatment.

Finally, the effect of chronic Cf-hGP administration on aging-induced cognitive impairment—specifically spatial memory—was assessed by the Morris water maze, which is typically used to test hippocampal function [52]. Four weeks after the oral administration of Cf-hGP to older rats (12 months old), acquisition training to learn the platform position was performed for three days. Four trials per day were performed to measure the ability of rats to learn the platform place, and the latencies on the third day were decreased by up to ˂30 s, indicating that memory had formed (Figure 6A,B). Following seven days of withdrawal, the probe trial was performed using a new starting position with (latency to platform) or without (crossing frequency) the submerged platform to verify whether the memory of the target location was maintained. Cf-hGP administration or the inhibition of ER stress with salubrinal (1 mg/kg, i.p.) significantly decreased the latency to reach the platform and increased the frequency of crossing the platform (Figure 6C,D). These data indicate that the ER stress-induced cognitive impairment that occurs with age could be attenuated with chronic Cf-hGP treatment.

Taken together, the aging-induced cognitive dysfunction caused by increased ER stress can be attenuated by activation of the MMP9-BDNF-ERK1/2 signaling pathway in the dorsal hippocampus by chronic Cf-hGP treatment (Figure 7). Considering the diverse functions of BDNF in neurogenesis and memory persistence, our results suggest that Cf-hGP treatment might have beneficial effects in controlling ER stress-induced cognitive dysfunction and neurodegenerative diseases associated with age.

## 4. Materials and Methods

### 4.1. Hydrophilic Glycoproteins from C. fulvescens

*C. fulvescens* was obtained from a farm at Wan-do island in Korea. *C. fulvescens* powder (20 g) was mixed with 1 L of sodium acetate (pH 6.0) and stirred overnight at 4 °C. The mixture was filtered using gauze and centrifuged at 5000 rpm for 10 min at 4 °C. The supernatant was transferred to a fresh 250-mL bottle and mixed at a ratio 1:4:1:3 of supernatant: methanol: chloroform: distilled water, and the mixture was centrifuged at 12,000 rpm for 15 min at 4 °C. The supernatant above the interface was discarded. The remaining mixture was washed by adding three volumes of methanol and then centrifuging at 12,000 rpm for 15 min at 4 °C. The pellets were washed with methanol, resuspended in distilled water, transferred to a 50-mL Corning tube, and freeze-dried overnight. To isolate hydrophilic glycoproteins for oral administration, the freeze-dried powder was rehydrated with saline (0.9% NaCl) and centrifuged at 13,000 rpm for 30 min at 4 °C. The supernatants were purified using lectin wheat germ agglutinin resin (Thermo Fisher Scientific, Rockford, IL, USA).

### 4.2. Oral Administration of Cf-hGP

Aging (12 months; 650–800 g) and young (10 weeks; 250–300 g) male Sprague-Dawley rats were obtained from Samtako Inc. (Gyeonggi, Korea). Rats (*n* = 6–8 per group) were housed in pairs in a controlled environment and maintained on a 12-h light/dark cycle during all of the experimental treatments. Food and water were provided ad libitum. All of the animal experiments were approved by the Animal Ethics Committee of Pukyong National University (approval no. 2018-25) and performed in accordance with the guidelines for the Care and Use of Laboratory Animals. Oral administration schemes for Cf-hGP (15 mg/kg/day) or saline are illustrated in Figure 2C. Briefly, the Cf-hGP was orally administered once per day for four weeks. Following the last administration, rats were subjected to acquisition training for three days with four trials per day. After seven days of withdrawal, spatial learning memory was assessed based on escape latency.

### 4.3. Preparation of Crude Synaptosomal Fraction

The dorsal hippocampus was removed after the aging rats (12 months, 650 to 800 g) were deeply anesthetized with a mixture of Zoletil 50 (18.75 mg/kg; Virbac, Seoul, Korea) and Rompun (5.83 mg/kg; Bayer, Anseong, Korea). Sections were serially cut using a microtome (Microm HM 430; MICROM International GmbH, Dreieich, Germany), and the dorsal hippocampus was removed using a steel borer (inner diameter, 2 mm). The lysis of hippocampal tissue samples was performed using cold lysis buffer (pH 7.5) containing 0.32 M of sucrose, 5 mM of 4-(2-hydroxyethyl)-1-piperazineethanesulfonic acid, and protease inhibitor cocktail (Thermo Fisher Scientific). After centrifugation at 1000× *g* for 10 min at 4 °C to separate nuclei, the supernatant was transferred to a new 1.5-mL tube and centrifuged at 12,000× *g* for 10 min at 4 °C. The supernatant was removed, and the pellet containing crude synaptosome was used in this study [53].

### 4.4. Immunoblotting

Protein concentrations were determined using a bicinchoninic acid protein assay kit (Thermo Fisher Scientific), and proteins (20 µg) were separated by 12% SDS-PAGE. Separated proteins were transferred to polyvinylidene difluoride membrane. The membrane was blocked with a blocking buffer containing Tris-buffered saline, 1% bovine serum albumin, and 0.1% Tween 20, and then probed with primary antibodies for BDNF (Abcam, Cambridge, MA, USA), GRP78, JNK, phosphor (p)-JNK, glycogen synthase kinase-3 beta (GSK-3β), pGSK-3β (Tyr216), pGSK-3β (Ser9), matrix metallopeptidase 9 (MMP9), and β-tubulin (1:1000; Cell Signaling Technology, Danvers, MA, USA) overnight at 4 °C on a shaker. After washing three times with Tris-buffered saline and 0.1% Tween 20 for 10 min, membranes were incubated with the corresponding secondary antibody (Thermo Fisher Scientific) at a dilution of 1:10,000 for 60 min at room temperature. The membrane was stripped and reprobed with anti-β-tubulin antibody to normalize the blots.

### 4.5. Double-Immunofluorescence Staining

Double-immunostaining was performed to confirm the expression of GRP78 in the rat dorsal hippocampus as previously described [54]. Following two washes with Dulbecco’s phosphate-buffered saline (DPBS) with Ca^2+^ and Mg^2+^, brain sections (30-µm thickness) were fixed with 4% paraformaldehyde for 20 min and permeabilized with 0.3% Triton X-100 (diluted in DPBS with Ca^2+^ and Mg^2+^) for 5 min at room temperature. After three washes with PBS, the sections were incubated for 60 min at room temperature with a 5% goat serum solution in DPBS with Ca^2+^ and Mg^2+^, and then incubated overnight at 4 °C in a mixture of rabbit anti-GRP78 and mouse anti-neuronal nuclear antigen primary antibodies (dilution, 1:500). After washing three times with DPBS with Ca^2+^ and Mg^2+^, the cells were incubated in a mixture of two secondary antibodies (goat anti-rabbit IgG-Alexa Fluor 488 and goat anti-mouse IgG-Alexa Fluor 647; both 1:500) for 60 min at room temperature. Following three washes and staining with 4’,6-diamidino-2-phenylindole solution for 10 min, the cells were mounted with a drop of ProLong gold anti-fade reagent (Gibco, Grand Island, NY, USA). Antibodies and normal goat serum for double-immunostaining were purchased from Abcam. The fluorescent images were taken using an EVOS^®^ FL imaging system (Thermo Fisher Scientific).

### 4.6. Rat Surgery and Microinjection Procedure

Rats were deeply anesthetized with a mixture of Zoletil 50 (18.75 mg/kg; Virbac) and Rompun (5.83 mg/kg; Bayer) and placed in a stereotaxic apparatus (Stoelting Co., Wood Dale, IL, USA). Under aseptic conditions, an infusion guided cannula (22 gauge; Plastics One, Roanoke, VA, USA) was implanted into the cingulate cortex using the following coordinates from the bregma: anterior–posterior, −2.5 mm; dorsal–ventral, −4.5 mm; medial–lateral, 1 mm. After a 28-gauge dummy cannula was inserted to prevent the guide cannula from clogging, the rats were given at least 3 days to recover. Following replacement with a 28-gauge internal cannula that protruded 0.5 mm beyond the guide cannula, function-blocking anti-BDNF antibody (1 μg/μL; Abcam) or inhibitors (MMP9 inhibitor, MMP9 inhibitor 1; GSK-3β inhibitor, CHIR; JNK inhibitor, SP600125; ERK1/2 inhibitor, SL327; Tocris Bioscience, Minneapolis, MN, USA) diluted in artificial cerebrospinal fluid (aCSF) were infused into the dorsal hippocampus, or aCSF was infused into the right dorsal hippocampus using a 2-µL Hamilton micro-syringe (Reno, NV, USA) in freely moving rats 30 min prior to the last oral administration of Cf-hGP. After the microinjection, the internal cannula was left in place for an additional 5 min to prevent any possible backflow.

### 4.7. Morris Water Maze Test for Spatial Learning and Memory

Following the oral administration of Cf-hGP for 4 weeks, hippocampal-dependent learning and memory were assessed by the Morris water maze test using a stainless-steel tank (diameter, 120 cm and depth, 45 cm). The platform was submerged 1 cm below the surface, and the water temperature was maintained at 25 °C. A set of semirandom starting positions was selected for basic acquisition training with the platform located in the southwest quadrant. Learning trials were conducted for three days (four trials/day). Each trial was limited to 2 min, and the interval between trials was 15 s. Seven days after the last learning trial, reference memory was measured.

### 4.8. Statistics

Data were expressed as the mean ± standard error of the mean for each group. Statistically significant differences among groups were determined by one-way analysis of variance with repeated measures followed by Tukey’s post-hoc test using Prism 5 (GraphPad Software, San Diego, CA, USA). A *p*-value ˂ 0.05 was considered statistically significant.

## Figures and Tables

**Figure 1 marinedrugs-17-00168-f001:**
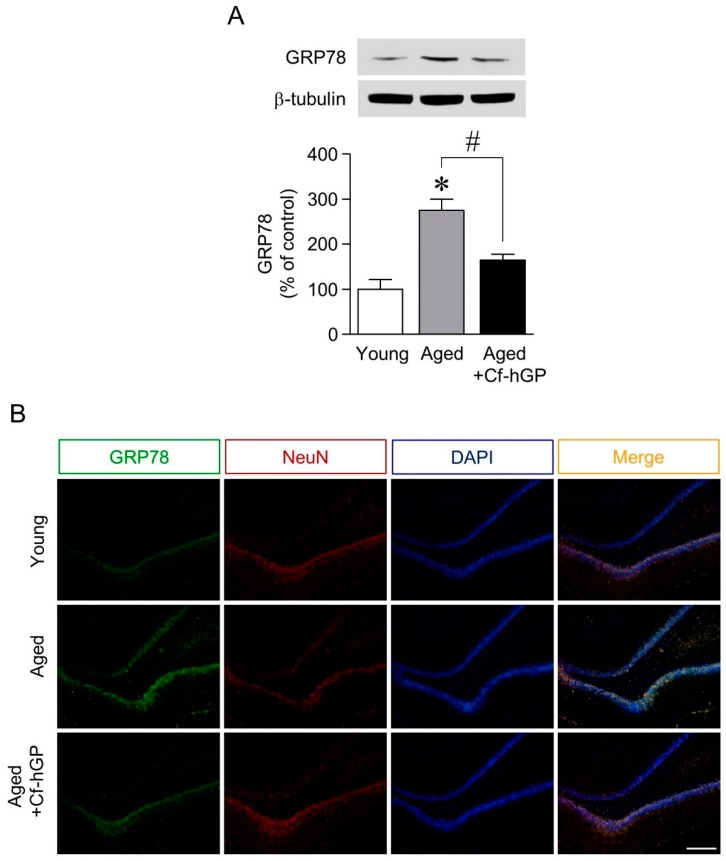
Effect of chronic oral administration of hydrophilic *C. fulvescens* glycoproteins (Cf-hGP) on glucose-regulated protein 78 (GRP78) expression in aged dorsal hippocampus. Expression of GRP78 was significantly increased in aged rat dorsal hippocampi, which was attenuated by chronic oral administration of Cf-hGP (15 mg/kg/day) for four weeks (**A**). The immunofluorescence levels of GRP78 in dorsal hippocampi were also downregulated by chronic Cf-hGP treatment (**B**). * *p* < 0.05 versus younger group; # *p* < 0.05 versus older group. Scale bar represents 200 µm. NeuN, neuronal nuclear antigen.

**Figure 2 marinedrugs-17-00168-f002:**
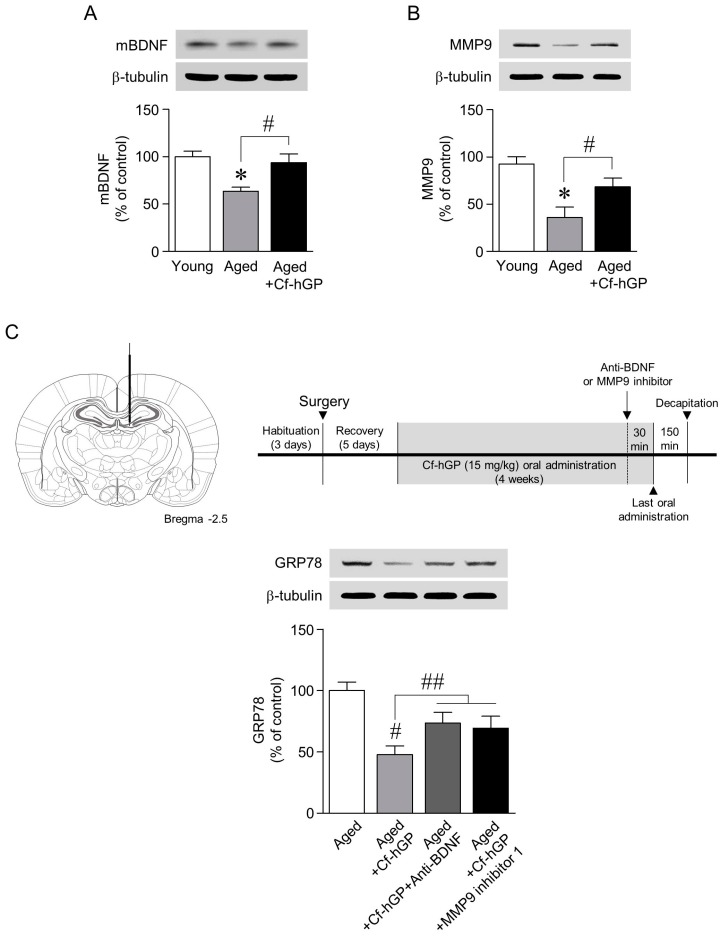
Regulation of mature brain-derived neurotrophic factor (BDNF) and matrix metallopeptidase 9 (MMP9) expression by chronic Cf-hGP administration and the involvement of aging-induced GRP78 expression in the dorsal hippocampus. The aging-induced decrease in mature BDNF and MMP9 expression in the dorsal hippocampus was restored by chronic Cf-hGP administration (**A**,**B**). Inhibition of BDNF or MMP9 signaling with microinjection of anti-BNDF antibody (1 μg/μL) or MMP9 inhibitor 1 (5 μM) proximal to the dentate gyrus of the dorsal hippocampus significantly downregulated the Cf-hGP-induced decrease in GRP78 expression (**C**). * *p* < 0.05 versus younger group; # *p* < 0.05 versus older group; ## *p* < 0.05 versus Cf-hGP-treated aging group.

**Figure 3 marinedrugs-17-00168-f003:**
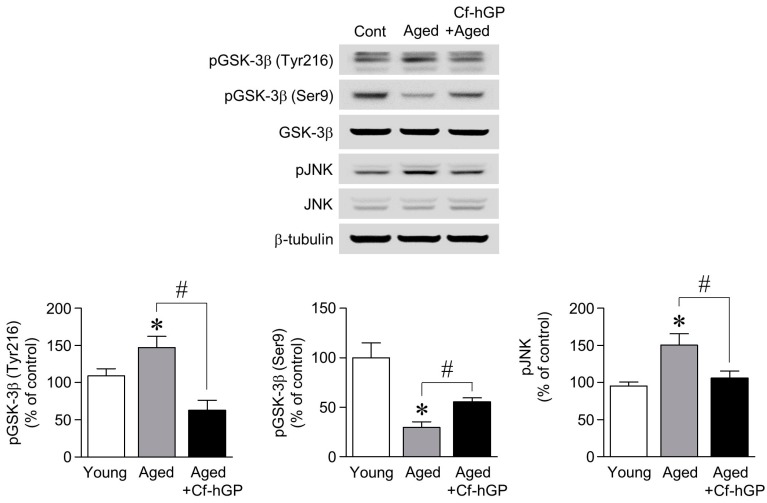
Phosphorylation of glycogen synthase kinase-3 beta (GSK-3β) and c-Jun N-terminal kinase (JNK) in the dorsal hippocampus of older rats by Cf-hGP administration. The phosphorylation of GSK-3β (Tyr216) and JNK in the dorsal hippocampi of older rats was significantly increased, while the phosphorylation of GSK-3β (Ser9) was significantly decreased; this effect was downregulated by chronic Cf-hGP administration. * *p* < 0.05 versus younger group; # *p* < 0.05 versus older group.

**Figure 4 marinedrugs-17-00168-f004:**
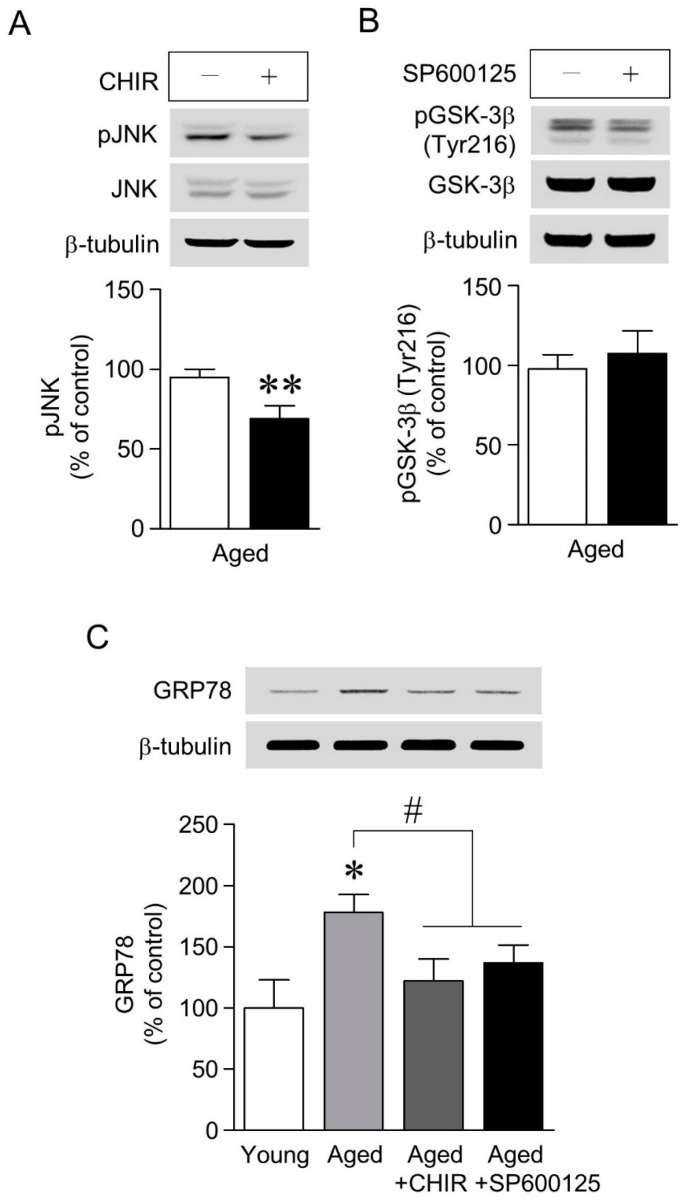
Aging-induced GRP78 expression caused by GSK-3β-mediated JNK signaling. Inhibition of GSK-3β phosphorylation with CHIR (3 μM) significantly decreased the aging-induced increase in JNK phosphorylation (**A**), but phosphorylation of GSK-3β (Tyr216) was not regulated by inhibition of JNK with SP600125 (10 µM), even if considering indirect effects (**B**). The increase in GRP78 expression with age was significantly downregulated by inhibition of GSK-3β or JNK with CHIR (3 μM) and SP600125 (10 µM), respectively (**C**). * *p* < 0.05 versus younger group; ** *p* < 0.05 versus no treatment of inhibitor; # *p* < 0.05 versus older group.

**Figure 5 marinedrugs-17-00168-f005:**
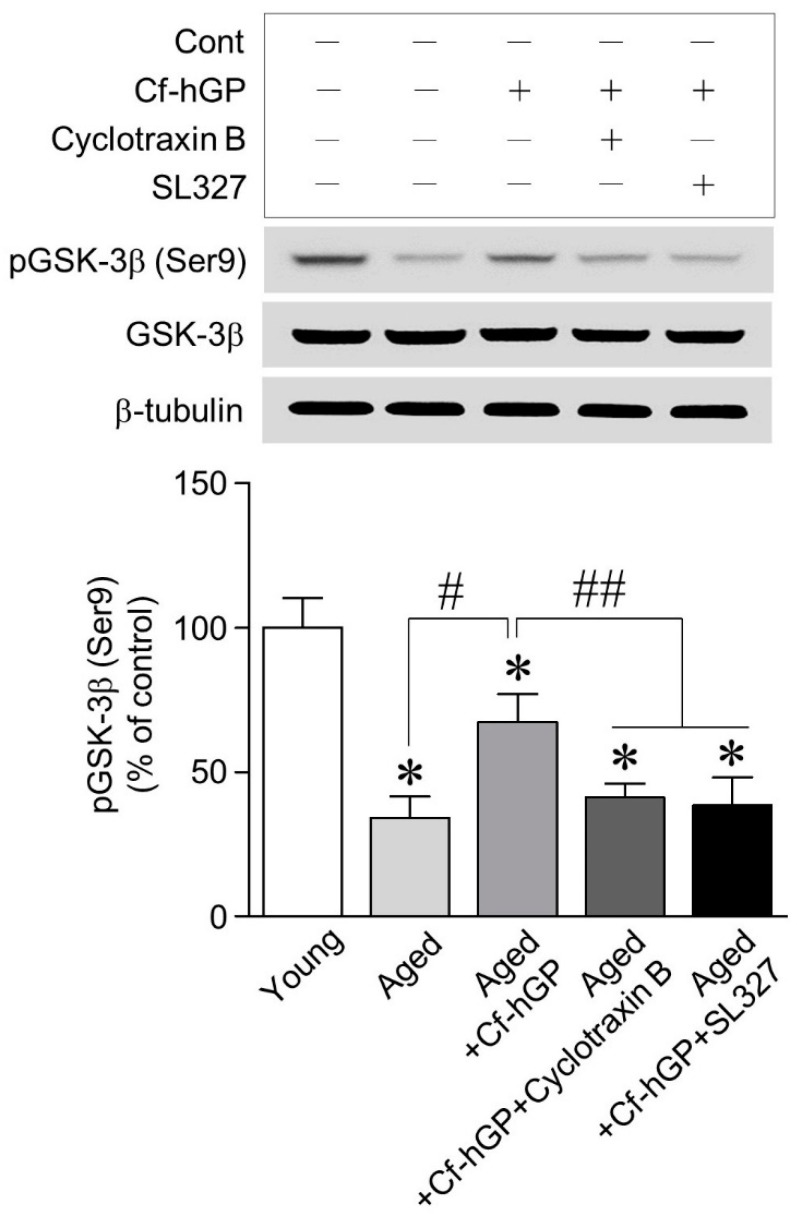
Phosphorylation of GSK-3β (Ser9) by Cf-hGP-induced synaptosomal extracellular signal-regulated kinase (ERK) 1/2 activation in the dorsal hippocampus. The phosphorylation of GSK-3β (Ser9) with age was significantly decreased; this effect was attenuated by chronic Cf-hGP administration. In addition, the Cf-hGP-induced increase in GSK-3β (Ser9) phosphorylation was downregulated by inhibiting TrkB and ERK1/2 with cyclotraxin-B (200 nM) and SL327 (10 μM), respectively. * *p* < 0.05 versus younger group; # *p* < 0.05 versus older group; ## *p* < 0.05 versus Cf-hGP-treated aging group.

**Figure 6 marinedrugs-17-00168-f006:**
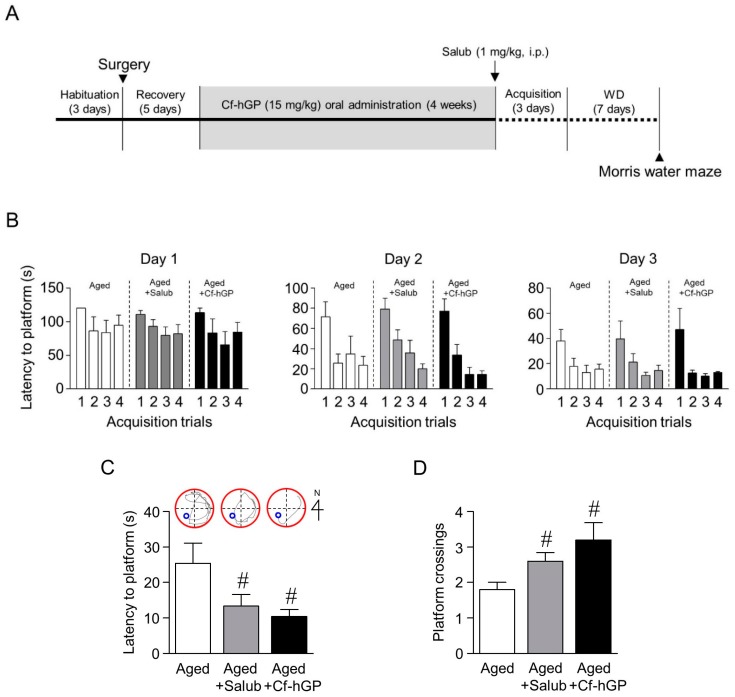
Effects of Cf-hGP on endoplasmic reticulum (ER) stress-induced cognitive impairment in aged rats. Following oral administration of Cf-hGP (15 mg/kg) once a day for four weeks, spatial memory was measured using the Morris water maze. The training phase was conducted for three days (**A**), and the spatial memory test was performed seven days after the last training trial (**B**). The latency to reach the platform was significantly decreased by Cf-hGP administration or inhibition of ER stress with salubrinal (1 mg/kg, i.p.) (**C**). The decrease in the aging-induced frequency of crossing the platform was also downregulated by Cf-hGP administration or the inhibition of ER stress (**D**). # *p* < 0.05 versus older group. Salub, salubrinal.

**Figure 7 marinedrugs-17-00168-f007:**
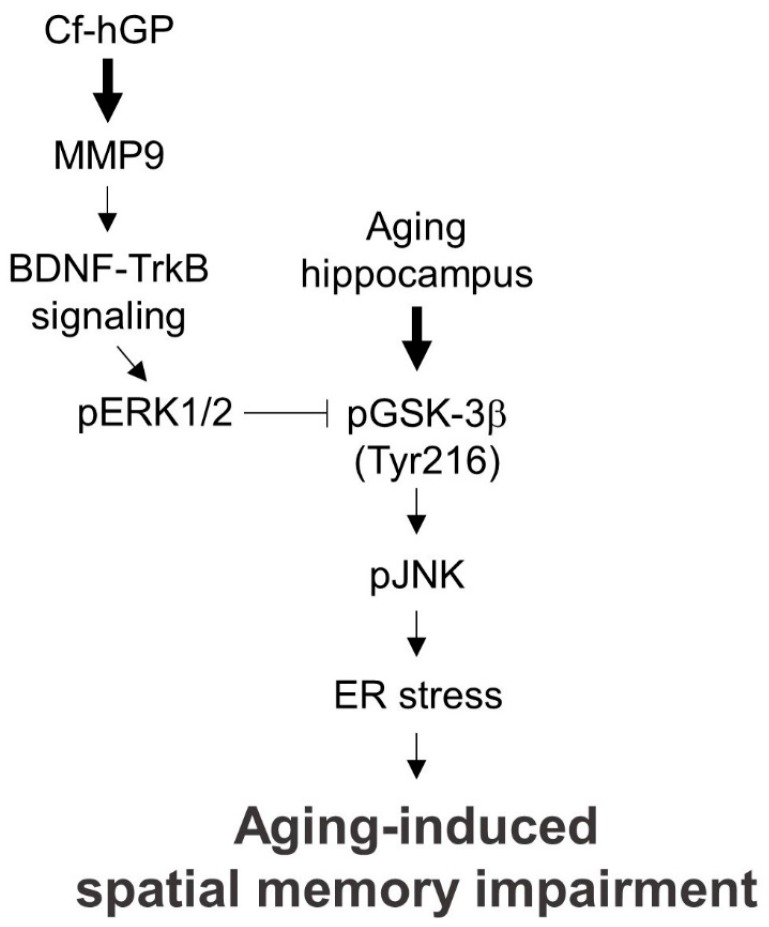
Schematic of a proposed mechanism underlying the effects of Cf-hGP against ER stress-induced spatial memory impairment in aged rat. Aging-induced spatial memory impairment results from increased ER stress in the dorsal hippocampus, which is downregulated by the chronic oral administration of Cf-hGP. Moreover, the increased ER stress caused by GSK-3β-mediated JNK signaling could be downregulated via the activation of synaptosomal BDNF–ERK1/2 signaling induced by the Cf-hGP treatment.

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
