# Peer review of "Hydrophilic Glycoproteins of an Edible Green Alga Capsosiphon fulvescens Prevent Aging-Induced Spatial Memory Impairment by Suppressing GSK-3β-Mediated ER Stress in Dorsal Hippocampus"

_marinedrugs, 2019, doi:10.3390/md17030168_

Reviewer 1 Report

This manuscript aims at using Capsosiphon fulvescens (C. fulvescens) crude proteins to enhance spatial memory by increasing the expression of brain-derived neurotrophic factor (BDNF) in rat dorsal hippocampus.

The paper is well-written and explained and the subject is interesting. However, some minor aspects can be improved:

Results and discussion:

1. Figure 1 (A) – The legend on the graphic is not visible, concerning the last bar. Please adjust the figure in conformity.

2. In some figures the names of the series are in the x axis and in other is on the side. Please uniformize this for the same in all figures.

3. Line 259 – Correct in the sentence “maze, which is. Typically…” For maze, which is typically…”.

 Materials and methods:

4. How many animals were used? In several figures in Results section is stated that authors used young rats, but they were not described in the MM section. Please correct this and the number used as well.

 References:

5. Correct reference number 53, the journal is in a different type of letter.

Author Response

Responses to Reviewers

 The authors wish to sincerely thank reviewers for their careful review and constructive criticism, which have been very useful for further improving the quality of this manuscript. In response to reviewers, we have revised the manuscript thoroughly according to reviewers' comments and changes in the revised manuscript are highlighted in red.
Reviewers' comments:

 Reviewer #1: This manuscript aims at using Capsosiphon fulvescens (C. fulvescens) crude proteins to enhance spatial memory by increasing the expression of brain-derived neurotrophic factor (BDNF) in rat dorsal hippocampus. The paper is well-written and explained and the subject is interesting. However, some minor aspects can be improved:

Results and discussion:

 1. Figure 1 (A) – The legend on the graphic is not visible, concerning the last bar. Please adjust the figure in conformity.

Response: We revised it.

2. In some figures the names of the series are in the x axis and in other is on the side. Please uniformize this for the same in all figures.

Response: We revised and unified the label of x axis of all figures.

3. Line 259 – Correct in the sentence “maze, which is. Typically…” For maze, which is typically…”.

Response: We misspelled and revised the sentence by deleting the period “.”.

 Materials and methods:

 4. How many animals were used? In several figures in Results section is stated that authors used young rats, but they were not described in the MM section. Please correct this and the number used as well.

Response: We revised the sentences line 297 to 299.

 References:

 5. Correct reference number 53, the journal is in a different type of letter.

Response: We revised it.

Reviewer 2 Report

In this manuscript, the authors evaluated the effects of C. fulvescens glycoproteins on aging-induced cognitive dysfunction and investigated their underlying mechanism of action. I agree aging-induced cognitive impairment is associated with ER stress caused by GSK-3β (Tyr216)-mediated JNK signaling, but which could be restored by Cf-hGP-induced activation of synaptic BDNF-ERK1/2 signaling seems less convincing due to lack of related experiments.

1)      GSK-3β (Tyr216) mediated JNK signaling is associated with the expression of GRP78, biomarker of ER stress, which has been proved in Section 2.1-2.4. So, In Section 2.5, I think it’s important to investigate if the GSK-3β (Tyr216) phosphorylation was affected by inhibition of the TrKB receptor and ERK1/2 signaling in order to link these two pathways.

2)      The authors proved that Cf-hGP increased the phosphorylation of GSK-3β(Ser9) by TrKB receptor and ERK1/2 activation (Section 2.3 and 2.5), which is meaningful, but as we all know, drugs have both efficacy and side effect, to prove if GSK-3β(Ser9) is on pathway, I think it’s necessary to investigate the relationship between phosphorylation of GSK-3β(Ser9) and the expression of GRP78, as shown in Fig 4.

3)      Line 19, add abbreviation after “matrix metallopeptidase 9 inhibitor”

4)      I think the authors could add some introduction about MMP9 and its relationship with BDNF in introduction part.

5)      Line 52, “Recent evidence indicates that BDNF reduces neuroplasticity and leads to spatial memory impairment in aging rat hippocampi” seems missing “deficiency” after BDNF.

In short, I think this manuscript could be considered for publication in Marine Drugs after major revisions.

Author Response

Responses to Reviewers

 The authors wish to sincerely thank reviewers for their careful review and constructive criticism, which have been very useful for further improving the quality of this manuscript. In response to reviewers, we have revised the manuscript thoroughly according to reviewers' comments and changes in the revised manuscript are highlighted in red.

Reviewers' comments:

 Reviewer #2: In this manuscript, the authors evaluated the effects of C. fulvescens glycoproteins on aging-induced cognitive dysfunction and investigated their underlying mechanism of action. I agree aging-induced cognitive impairment is associated with ER stress caused by GSK-3β (Tyr216)-mediated JNK signaling, but which could be restored by Cf-hGP-induced activation of synaptic BDNF-ERK1/2 signaling seems less convincing due to lack of related experiments.

1) GSK-3β (Tyr216) mediated JNK signaling is associated with the expression of GRP78, biomarker of ER stress, which has been proved in Section 2.1-2.4. So, In Section 2.5, I think it’s important to investigate if the GSK-3β (Tyr216) phosphorylation was affected by inhibition of the TrKB receptor and ERK1/2 signaling in order to link these two pathways.

Response: As you well known, ERK1/2 is a serine/threonine kinase, not tyrosine. Cf-hGP treatment significantly decreased tyrosine 216 phosphorylation of GSK-3β (Tyr216, active form) but increased serine 9 phosphorylation of GSK-3β (Ser9, inactive form). In addition, Cf-hGP-induced increase in GSK-3β (Ser9) phosphorylation was downregulated with TrkB receptor and ERK1/2 inhibition with cyclotraxin B and SL327, respectively. These data indicate that activation of TrkB-mediated ERK1/2 signaling induces inactivation of GSK-3β (Ser9 phosphorylation) and downregulates GSK-3β-mediated GRP78 expression.

Although it was not measured whether GSK-3β (Tyr216) phosphorylation is directly regulated by ERK1/2, it seems to be enough to show that the increase in ERK1/2-mediated inactivation of GSK3β by Cf-hGP treatment downregulates the GRP78 expression caused by GSK-3β activation (Tyr216 phosphorylation). These are described in detail in discussion part from lines 244 to 254.

2) The authors proved that Cf-hGP increased the phosphorylation of GSK-3β(Ser9) by TrKB receptor and ERK1/2 activation (Section 2.3 and 2.5), which is meaningful, but as we all know, drugs have both efficacy and side effect, to prove if GSK-3β(Ser9) is on pathway, I think it’s necessary to investigate the relationship between phosphorylation of GSK-3β(Ser9) and the expression of GRP78, as shown in Fig 4.

Response: As mentioned in the response #1, the serine 9 phosphorylation is an inactive form of GSK-3β. In addition, aging-induced increase in GRP78 expression was associated with JNK phosphorylation mediated by GSK-3β activation. Thus, the increase in inactivation of GSK-3β (Ser9 phosphorylation) by Cf-hGP treatment induces decrease in JNK phosphorylation, suggesting that the JNK-mediated GRP78 expression is downregulated by GSK-3β (Ser9) phosphorylation.     

3) Line 19, add abbreviation after “matrix metallopeptidase 9 inhibitor”

Response: The abbreviation of matrix metallopeptidase 9 inhibitor 1 is “MMP9 inhibitor 1”. Because matrix metallopeptidase 9 was mentioned once in the abstract, we did not use the abbreviation “MMP9”.   

4) I think the authors could add some introduction about MMP9 and its relationship with BDNF in introduction part.

Response: We investigated the effect of Cf-hGP on aging-induced cognitive dysfunction focusing BDNF-mediated ER stress regulation. To clarify research objectives, thus, a signal regulator such as MMP9 and its relationship with BDNF was described in detail in discussion part line 216 to 221. We appreciate your understanding in the matter.  

5) Line 52, “Recent evidence indicates that BDNF reduces neuroplasticity and leads to spatial memory impairment in aging rat hippocampi” seems missing “deficiency” after BDNF.

 In short, I think this manuscript could be considered for publication in Marine Drugs after major revisions.

Response: We misspelled and revised the sentence. 

Round  2

Reviewer 2 Report

The revised manuscript can be accepted in present form.